# The Relationship Research between Biodiversity Conservation and Economic Growth: From Multi-Level Attempts to Key Development

**Yutong Zhang [1], Wei Zhou [2]** and **Danxue Luo [2,*]**

[1] School of Business and Tourism Management, Yunnan University, Kunming 650091, China
[2] Business School, Yunnan University of Finance and Economics, Kunming 650221, China
* Correspondence: luodanxue@stu.ynufe.edu.cn

**Abstract:** The relationship between biodiversity and economic growth is a topic that still needs to be considered in a volatile global environment. Therefore, a bibliometric analysis of this topic can help scholars understand the current state of research and topical issues. Based on CiteSpace and Pajek, this paper fully does this study from the perspectives of authors, journals, countries, keywords, regions, and path analysis. Through this research, we find that: (1) the number of publications and citations in the literature about biodiversity and economic growth research have increased significantly; (2) scholars oppose unrestricted economic growth and advocate for the protection of the environment and biodiversity. Ecological environment protection and economic development are win-win relationships. (3) The keyword analysis revealed that a current research hotspot is the question of how to develop the economy while preserving ecological diversity. (4) Developed countries or regions are pioneers in studying the relationship between biodiversity and economic growth, and they have progressively led and driven the development of research in this field. The main purpose of this study is to provide guidance to researchers, and those interested in biodiversity and economic growth.

**Keywords:** biodiversity conservation; economic growth; environmental protection; treatment and supervision; bibliometrics analysis

## 1. Introduction

The problems caused by the dramatic decline of biodiversity have attracted worldwide attention and reflection [1]. One of the main concerns is how to balance the relationship between biodiversity and economic growth [2]. Economic growth will inevitably require the sacrifice of some natural resources. Additionally, the drastic decline in biodiversity will threaten the survival and development of human beings, destroy the future source of food and medicine for human beings, and even affect the ecological balance of the whole of nature [3]. Therefore, it is of great value and significance to study the relationship between biodiversity and economic growth.

"Biological diversity" (biodiversity) is the term that first appeared in nature conservation journals in the 1980s [4]. Biodiversity refers to the diversity among all organisms, including terrestrial, marine, and other aquatic ecosystems, and the ecological complexes of which they are composed, including intra-species, inter-species, and ecosystem diversity. Earlier, papers related to biodiversity and economic growth focused on the relationship between the natural environment and people [5]. Sustained economic growth drives industrial expansion and accelerates communication and trade, which leads to the excessive consumption of materials [6] and energy [7]. The proliferation of fuel use has led to a significant increase in greenhouse gas emissions [8] and climate change [9], with significant negative impacts on biodiversity [10]. Specifically, economic growth affects biodiversity in

three ways: environmental pollution, land misuse, and climate change [11]. Researchers have developed studies around these aspects.

The research literature concerning economic growth and environmental pollution can be traced back to two studies in the early 1990s, which were Grossman and Krueger (1992) [12] and the World Bank Development Report of 1992 [13]. The report also pointed out that today's environmental problems and economic growth show an "inverted U-shaped" curve relationship. As the economy grows, per capita income increases, and biodiversity destruction increases. However, this phenomenon will be improved and mitigated when economic growth reaches a certain level. In the literature, the "inverted U-shaped" curve between environmental pollution and economic growth is also known as the environmental Kuznets curve (EKC) [14,15]. The consensus of relevant studies is that environmental quality deteriorates in the early stages of economic development/growth and gradually improves in the later stages [16]. Numerous studies have been conducted to examine the relationship between economic growth and environmental pollution from theoretical and empirical perspectives in the following two decades.

Land misuse, such as agricultural expansion, and urban and infrastructure development, can encroach and fragment habitats of organisms, thus causing biodiversity loss [17,18]. Meanwhile, intensification of agricultural techniques can reduce soil organic matter, disturb soil biomes [19], increase the risk of soil erosion, degradation [20,21], and salinization [22], lead to biohomogenization, and be toxic to plants [23], and threaten birds, mammals, amphibians, and insects [24,25]. Ecosystems and many species were influenced by global climate, changing migration trajectories [26,27], living habits [28,29], and even some species on the verge of extinction [30]. In addition, climate change has increased the frequency and intensity of extreme events [31], such as droughts, floods, storms, etc. These kinds of extreme events pose a greater threat to biodiversity than global warming [32]. The effects of climate change may act synergistically with the effects of land use change [33].

In addition, governments, as well as scientists, are actively looking for ways to conserve biodiversity and transform it while trying to find out how economic growth affects it. Refs. [1,34] suggest that technology may be the lubricant between them. Recent related studies have also combined COVID-19 with biodiversity issues to discuss how the impact of the COVID-19 pandemic on economic growth extends to global biodiversity and its conservation [35–37]. Although the specific impacts of economic growth on biodiversity have been widely discussed, a balanced relationship between economic growth and biodiversity has yet to be constructed. Therefore, it is important to sort out and analyze the literature in this field. This work will help scholars quickly understand the current research status, gaps, and hot issues at the intersection of biodiversity and economic growth research. For the above purpose, we analyze the research trends and research hotspots on economic growth and biodiversity in terms of keywords, authors, institutions, journals, and countries with the help of tools, such as CiteSpace [38] and Pajek, based on the literature of the last two decades. Through previous studies, the characteristics and commonalities of the field are summarized, and directions and predictions are provided for subsequent research.

Commonly used tools in literature measurement are CiteSpace, Pajek, VOS viewer, Ucinet, Net Draw, etc. Their main purpose is to achieve cluster analysis on the relationship matrix of knowledge units in scientific literature and to form a clustering network. In this paper, two bibliometric tools, CiteSpace and Pajek, are mainly used. CiteSpace and Pajek are more suitable for the analysis of dynamics, complexity, and stages. The tool presents visualization results that help researchers examine the development of biodiversity and economic growth research through diverse insights. CiteSpace provides a cross-sectional overview of biodiversity research through a visual flow of information. Compared with other bibliometric methods, the research conducted in this paper through CiteSpace can present the results more intuitively. The literature clustering, author, journal, country, and institutional analysis included in CiteSpace can be used to conduct research from multiple dimensions. The interpretation of the literature data will clarify the frontiers of biodiversity development, research trends, and future research directions. CiteSpace has recently been

cited in more areas, such as healthcare. Ref. [39] used CiteSpace 5.3 R4 as a tool to assess the origins, current trends, and research hotspots of SCH during pregnancy. Pajek sorts out the development context and path of the literature. The analysis of the forward main path, global key main path, global standard main path, and backward main path can help to sort out the development of literature on biodiversity and economic growth [40]. In the study of [40], some analysis and visualization methods were introduced to implement large-scale networks in Pajek. More recently, Pajek has been used in areas such as clinical judgment for additional developmental screening and research into developmental monitoring processes [41]. CiteSpace and Pajek's research methods have also been applied to the healthcare field in recent years. Due to the outbreak of the new crown epidemic, more attention has been paid to the latest research and progress in the direction of healthcare.

The rest of this paper is organized as follows. Section 2 introduces the purpose and significance of the research and the current research trends. Section 3 combines the bibliometric tools of CiteSpace to cluster the literature on biodiversity and economic growth, authors, journals, countries, and institutions to make a comprehensive analysis. Section 4 uses Pajek to sort out the paths of the research topics and analyzes the research trends from the forward main path, the global key main path, the global standard main path, and the backward main path analysis. In the arrangement, Section 5 makes a summary of this paper and lists the relevant research conclusions.

## 2. Basic Preparations

### 2.1. Preliminary Analysis of Research on Biodiversity and Economic Growth

This research reveals the status and development trend of biodiversity and economic growth through the analysis in the field of cartography. Different analysis methods should be given by various and related software. In this paper, CiteSpace is used to analyze the research status of biodiversity and economic growth. Specifically, the year, author, institution, country, and frequently occurring keywords involved in the study were selected for analysis, and key information was highlighted.

This paper selects the Web of Science (WOS) as the bibliographic database, which includes the Science Citation Index Expanded (SCI-EXPANDED) and the Social Sciences Citation Index (SSCI). Because we chose WOS as the database, we 'do not need to transform it when we use CiteSpace for analysis, and it is a reputable and comprehensive database. Next, we set a data retrieval strategy:

Topic = "Biodiversity" and "Economic growth"
Timespan = "All year"
Databases = Science Citation Index Expanded (SCI-E) and Social Sciences Citation Index (SSCI)
Lemmatization = On

A total of 1606 papers related to biodiversity and economic growth were published from 1991 to November 2021. To further guarantee the reliability and unbiasedness of the results, we screened and refined the acquired data in the following three aspects: (1) we manually read the abstracts and keywords of all articles, assessed the relevance of the articles to the research topic, and eliminated irrelevant literature. (2) We refined the 'author's information, e.g., incomplete abbreviations, and excluded literature where the author was anonymous. (3) We checked the structure of all articles for completeness and eliminated documents with incomplete elements. Eventually, we obtained 1587 complete and valid documents.

According to the processed data, the number of publications per year is shown in Figure 1. Figure 1 shows the number of publications in the field of biodiversity and economic growth from 1991 to 2021, and the number of publications shows a continuously increasing trend. During the initial period of 1991 to 1993, little attention was paid to this field. However, after 1994, it can be seen that although the number of literary studies is not stable, it is also increasing year by year in a general trend. After 2018, there was a marked increase. Additionally, at this stage, the most cited literature mainly explored the contradiction between ecosystems and economic development [42]. Research at this stage

focused on solving practical problems between the destruction of ecological balance and economic development.

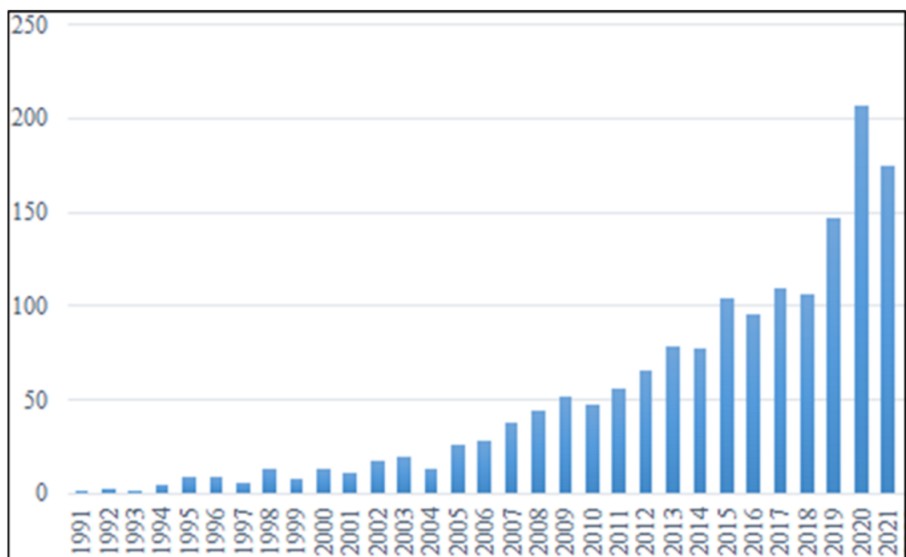

**Figure 1.** Number of publications from 1991 to 2021.

The development of research on economic growth and biodiversity can be divided into four stages: the flat period, fluctuation period, growth period, and explosion period. The flat period was from 1991 to 1993, when few scholars paid attention to this field and there was little literature. Countries around the world have made economic growth their top priority, ignoring its impact on and damage to the environment. Because economic growth will inevitably lead to ecological imbalances, the trade-off is that it will also be ignored [43].

The fluctuating period was from 1994 to 2004. The development of the literature during this decade was slightly above the plateau but was not a hot area of research for a large number of scholars. Based on the background of the time, countries were still in the stage of emphasizing economic development, so they would deliberately ignore the environmental problems brought by economic development.

The growth period was from 2005 to 2018. During this period, an increasing number of scholars have paid attention and the amount of published literature has gradually increased. Although the upward trend of the literature volume is not stable, it is also relatively flat in terms of time.

The outbreak period is from 2019 to 2021. With the yearly increase in natural disasters and the economic development of most countries, a large body of literature has emerged in this field. It indicates that when economic development reaches a certain level, people will pay more attention to the harmonious coexistence between humans and nature. Meanwhile, the outbreak of the COVID-19 epidemic has triggered more scholars to think about biodiversity and economic growth, further enriching the research literature in this area.

In the related literature on biodiversity and economic growth, the structure of the literature type is clearer and more objective than the structure of the research direction. It can be seen from Figure 2 that "environmental science" is the most concerned field, accounting for 34.00%, followed by "Ecology" which accounted for 22.33%, "environmental research" accounted for 16.84%, and "biodiversity conservation" accounted for 12.67%. The above fields determine the overall trend of research on biodiversity and economic growth.

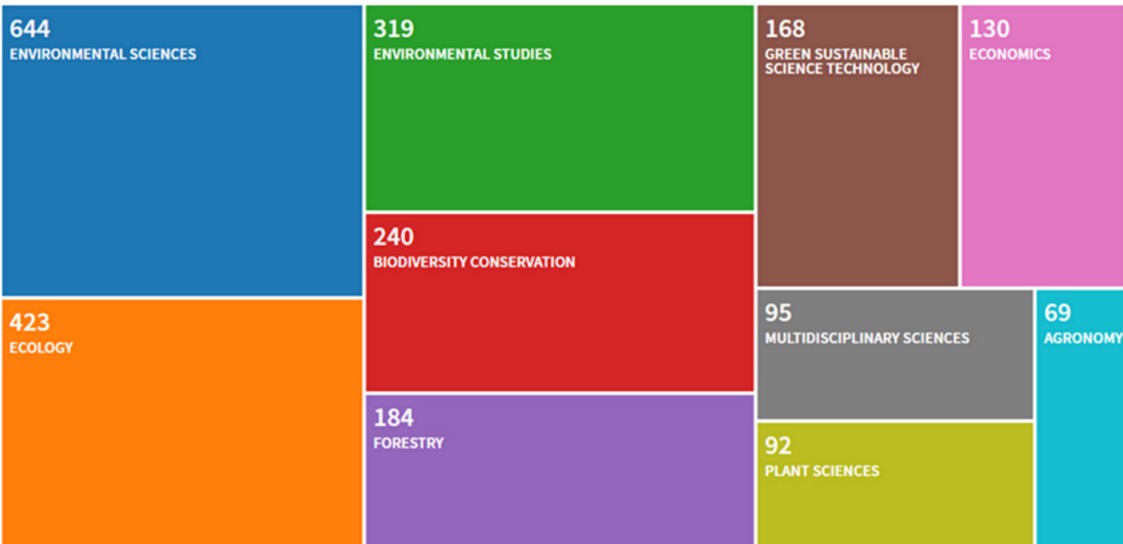

**Figure 2.** Types of literature on the relationship between biodiversity and economic development.

*2.2. Synergistic Analysis of the Literature Network on Biodiversity and Economic Growth*

To understand the theoretical underpinnings and trends in biodiversity and economic growth, we used CiteSpace to analyze literature data and examine the intrinsic connections between associated authors, institutions, and countries in the field.

Through the processing of data from CiteSpace, authors' collaborative networks can be identified, thereby making the relationship between different authors more accurate. Figure 3 shows the most cited authors in the area of biodiversity and economic development, with the number of citations proportional to tree ring size. Table 1 shows the top 10 highly productive authors in the field of biodiversity. The most frequently co-cited author was Fao (182), followed by World B (121), and Costanza R (121). The difference in co-cited frequency between authors ranked 4–6 was smaller, and the same occurred in the 7th–10th place. As shown in Figure 3, the 10 most prolific authors correspond to larger nodes and have more connections to other nodes, which indicates that they collaborate more frequently with other authors.

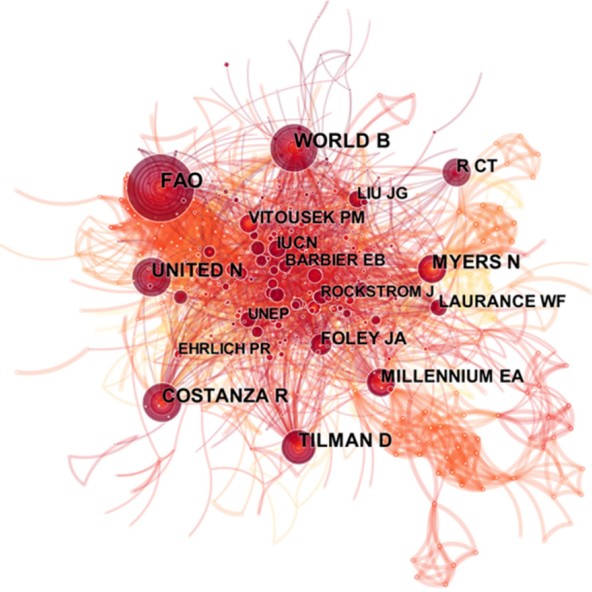

**Figure 3.** Literature citations in the field of biodiversity and economic development.

**Table 1.** Authors co-cited in the field of biodiversity.

| Cited Frequency | Year | Author | Cited Frequency | Year | Author |
|---|---|---|---|---|---|
| 182 | 2006 | Fao | 111 | 1999 | Tilman D. |
| 137 | 2000 | World B. | 90 | 2008 | Millennium E. A. |
| 121 | 1995 | Costanza R. | 80 | 2014 | R C. T. |
| 118 | 2005 | United N. | 78 | 2012 | Foley J. A. |
| 113 | 1995 | Myers N. | 71 | 2006 | Laurance W. F. |

*2.3. The Cooperation Analysis between Research Institutions and Countries*

Figure 4 shows the analysis of institutional collaboration, which measures how well institutions collaborate on research topics. The nodes in Figure 4 are closely connected to each other and spread in all directions, which indicates that the collaboration among institutions tends to mature.

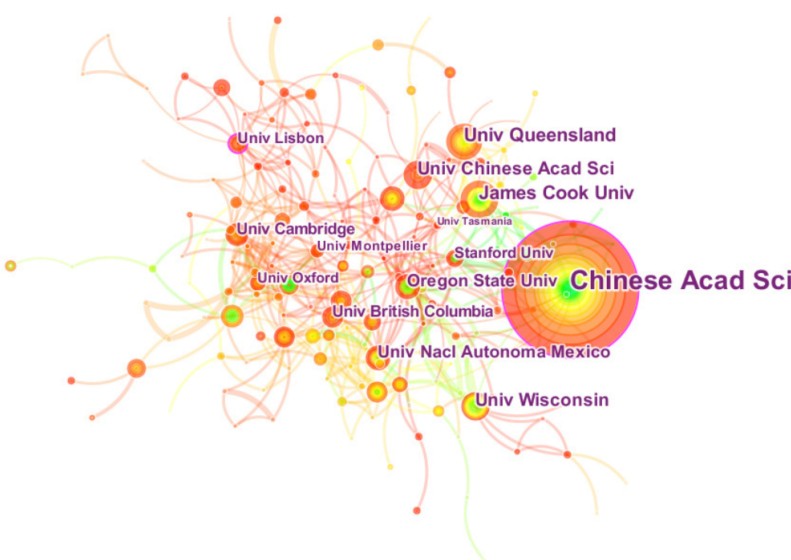

**Figure 4.** Institutional cooperation on biodiversity and economic growth.

From the Figure 4 institutional collaboration map, it can be seen that institutions with more research in this field are concentrated in China. It is undeniable that China's rapid economic development has directly led researchers to pay more attention to biodiversity and economic growth. Followed by Australia's James Cook University and the University of Queensland, it can be seen from the relevant Australian research institutions that follow, Australia, as a country with rich biodiversity, also has many policies related to ecological protection. Related studies and discussions provide advice and guidance on the balance between biodiversity conservation and economic development.

Table 2 lists the top ten high-yield institutions for biodiversity and economic growth research. As seen in Table 2, the Chinese Academy of Sciences published the most papers with 71, followed by James Cook University with 20, and the third-ranked institution was the University of Queensland with 19 publications. Numerous developed countries are important contributors to biodiversity research. It is worth mentioning that the Chinese Academy of Sciences has the highest centrality, which shows that it plays an important role in the institutional cooperation network. Because centrality is an indicator of the importance of nodes in the network.

**Table 2.** Ranking of research institutions on biodiversity and economic growth.

| Count | Centrality | Year | Institutions |
|:---:|:---:|:---:|:---:|
| 71 | 0.17 | 2001 | Chinese Academy of Sciences |
| 20 | 0.06 | 2008 | James Cook University |
| 19 | 0.05 | 2003 | University Queensland |
| 15 | 0.05 | 2019 | University of Chinese Academy of Sciences |
| 15 | 0.08 | 1998 | University of Wisconsin |
| 13 | 0.07 | 2011 | Universidad Nacional Autonoma de México |
| 13 | 0.05 | 2011 | Wildlife Conservation Society |
| 13 | 0.08 | 2008 | Oregon State University |
| 12 | 0.06 | 2007 | INRA |
| 12 | 0.09 | 2009 | University of Cambridge |

The data in Figure 5 mainly reflects the national or regional collaborative research on biodiversity and economic growth. As can be seen in Figure 5, the United States is the center of dissemination to neighboring countries. In addition, the connection between the two nodes, the U.S. and China, further illustrates the need for greater cross-border cooperation on this global issue. The collaboration of each country is shown in Table 3. The U.S. ranks first with 444 collaboration frequencies, followed by China and Australia with 190 and 161 collaboration frequencies, respectively. However, the countries or regions with relatively high centrality are France (0.12), Italy (0.09), and Germany (0.08).

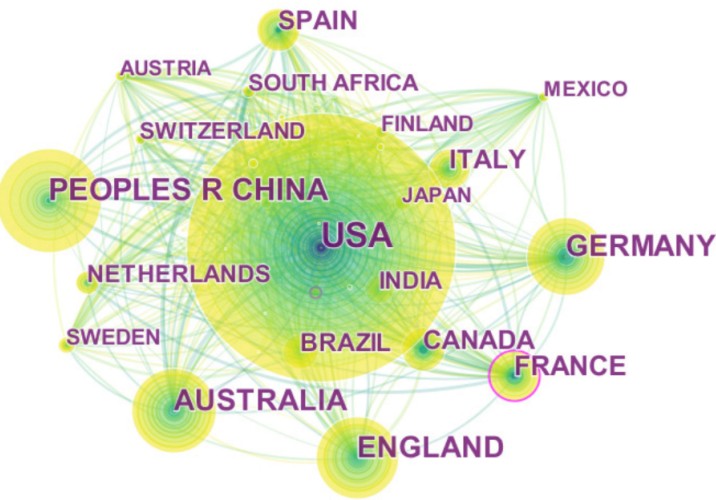

**Figure 5.** National and regional collaboration on biodiversity and economic growth.

As shown in Table 3, in previous research results, many developed countries and regions are important contributors to biodiversity research. The United States, for example, has far more research than any other country or region and it maintains close ties with national institutions that conduct research in related fields. This shows that when economic development reaches a certain level, developed countries will naturally return to attach importance to biodiversity conservation and environmental issues and balance the relationship between ecology and economic development. From Figure 5, we can also understand that regardless of the number of research articles published by countries and regions, they maintain close cooperation. As a globalization issue, developed countries have passed the stage where they need to pay too much attention to economic growth issues. Returning to the issue of ecological balance, developing countries are still the primary task for economic growth, so they do not pay too much attention to the balance and protection of biodiversity.

**Table 3.** Ranking of national and regional collaborations in the field of biodiversity and economic growth.

| Count | Centrality | Countries |
|-------|-----------|-----------|
| 444 | 0.08 | USA |
| 190 | 0.01 | China |
| 161 | 0.08 | Australia |
| 157 | 0.05 | England |
| 144 | 0.09 | Germany |
| 95 | 0.12 | France |
| 90 | 0.08 | Canada |
| 84 | 0.06 | Spain |
| 82 | 0.1 | Italy |
| 64 | 0.03 | Brazil |

*2.4. Keyword Analysis*

Figure 6 and Table 4 show the relatively intensive research words and phrases in the existing research: the first is biodiversity, a total of 663 times; the second is conservation, a total of 310 times; the third is growth, a total of 247 times; the fourth is management, a total of 223 times; the fifth is impact, with a total of 198 times; the sixth is ecosystem services, a total of 168 times; the seventh is climate change, a total of 167 times; the eighth is economic growth, a total of 144 times; the ninth is diversity, a total of 132 times; and the tenth is sustainability, a total of 111 times.

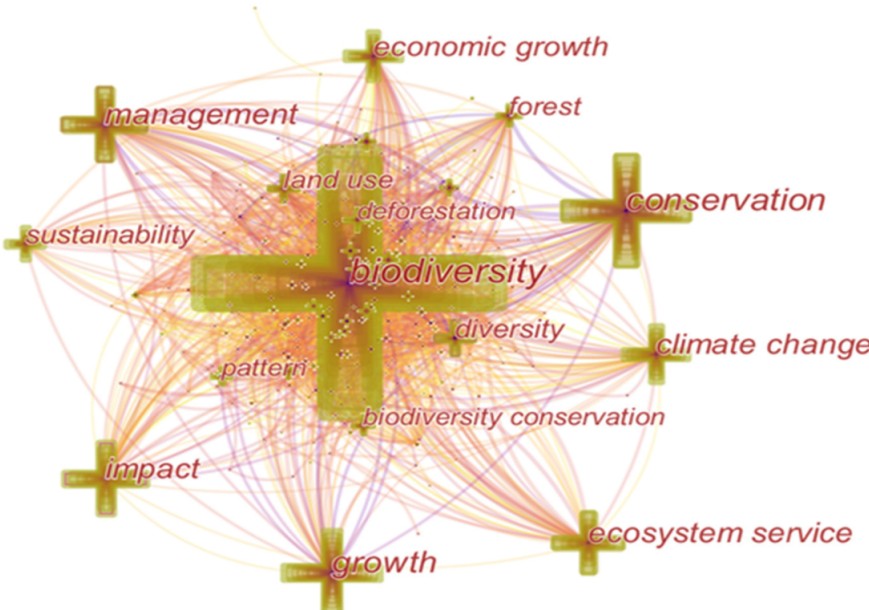

**Figure 6.** Keyword analysis of existing research on biodiversity and economic growth.

The ranking of the top ten keywords reveals that most of the literature studies focus on the construction of ecological balance rather than economic development. It indicates that economic development is no longer the top priority for most developed countries. Social groups will focus on the construction of an ecological civilization when economic development reaches a certain level.

**Table 4.** Keyword analysis ranking of studies related to biodiversity and economic growth.

| Count | Centrality | Year | Keywords |
|---|---|---|---|
| 663 | 0.07 | 1995 | biodiversity |
| 310 | 0.07 | 1995 | conservation |
| 247 | 0.05 | 1998 | growth |
| 223 | 0.12 | 1998 | management |
| 198 | 0.11 | 2003 | impact |
| 168 | 0.08 | 2006 | ecosystem service |
| 167 | 0.07 | 2004 | climate change |
| 144 | 0.06 | 2002 | economic growth |
| 132 | 0.11 | 1996 | diversity |
| 111 | 0.04 | 2008 | sustainability |

*2.5. Journal and Author Collaboration Networks*

In Table 5, among all the journals that publish biodiversity, the most cited journals are *Science*, with a total of 915 citations, followed by *Nature*, with a total of 756 citations, *Proceedings of the National Academy of Sciences of the United States of America*, with a total of 649 citations, and *Conservation Biology*, with a total of 570 citations. The collaborative network among journals is shown in Figure 7.

**Table 5.** Ranking of collaboration journals on biodiversity and economic growth.

| Cited Frequency | Centrality | Year | Journal |
|---|---|---|---|
| 915 | 0.11 | 1995 | *Science* |
| 756 | 0.17 | 1995 | *Nature* |
| 649 | 0.1 | 1999 | *Proceedings of the National Academy of Sciences of the United States of America* |
| 570 | 0.02 | 1995 | *Conservation Biology* |
| 546 | 0.07 | 1996 | *Biological Conservation* |
| 518 | 0.09 | 1998 | *Ecological Economics* |
| 460 | 0.05 | 1995 | *Bioscience* |
| 458 | 0 | 2009 | *Plos One* |
| 398 | 0.06 | 1999 | *Ecological Applications* |
| 379 | 0.04 | 1995 | *Trends in Ecology & Evolution* |

As shown in Figure 8 and Table 6, there is an increasing number of scholars focusing on biodiversity topics, each one analyzing the research from a different perspective. The top ten authors are listed in Table 6, and the data show that scholars represented by Radeloff V.C. occupy an important position in the collaborative network of authors, while they also maintain communication with other authors.

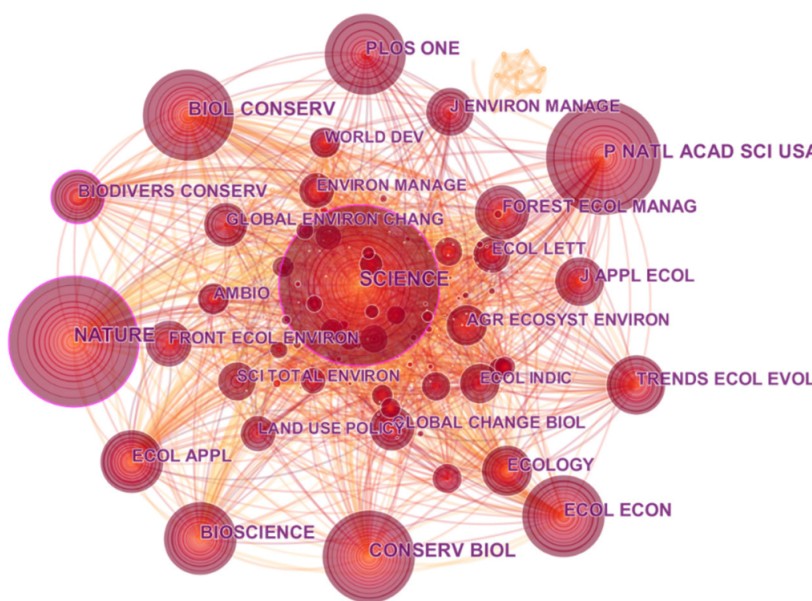

**Figure 7.** Co-citation network of biodiversity and economic growth journals.

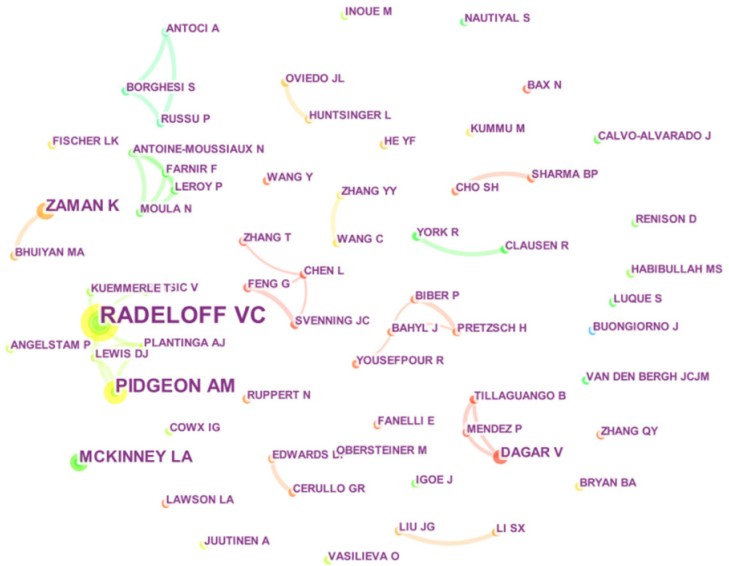

**Figure 8.** Analysis of author collaboration networks on biodiversity and economic growth.

Based on the above basic analysis of bibliometrics, it is possible to draw the following conclusions. With the attention of people on biodiversity, the number of relevant literature publications has increased year by year, and there are more related studies that can support bibliometrics. The reasons for the increase in the number of publications are as follows: first, the close cooperation between authors and institutions has led to an increase in the number of publications in related fields year by year. Second, taking China as an example, developing countries have also begun to protect the ecological environment under a development model in which economic growth is the main goal. In recent years, corresponding measures, such as returning farmland to forests, have been taken to establish a biodiverse environment under the premise of stable economic growth. Third, the cooperation between countries and institutions is concentrated in developed countries, which can also indicate that developed countries have generally passed the stage of economic development, thus turning the focus to biodiversity conservation.

**Table 6.** Ranking of author collaboration networks for research on biodiversity and economic growth.

| Count | Centrality | Year | Authors |
|---|---|---|---|
| 8 | 0 | 2012 | Radeloff V. C. |
| 5 | 0 | 2013 | Pidgeon A. M. |
| 4 | 0 | 2016 | Zaman K. |
| 4 | 0 | 2009 | Mckinney L. A. |
| 3 | 0 | 2021 | Dagar V. |
| 2 | 0 | 2020 | Bahyl J. |
| 2 | 0 | 2005 | Russu P. |
| 2 | 0 | 2010 | Igoe J. |
| 2 | 0 | 2019 | Cerullo G. R. |
| 2 | 0 | 2017 | He Y. F. |

## 3. Biodiversity and Economic Growth Co-Citation and Burst Detection

### 3.1. Literature Cluster Analysis on Biodiversity and Economic Growth

Table 7 shows the clustering of documents in biodiversity and economic growth research whose index terms were extracted from the citations of the studies and summarized. The size indicates the number of publications in the cluster. The largest cluster is cluster 0, with 76 constituent units, followed by cluster 1, which has 72 constituent units. The silhouette represents the average contour value of the cluster. It is generally considered that silhouette >0.5 is reasonable, and silhouette >0.7 means that the cluster is convincing. The higher the silhouette scores, the better the quality.

**Table 7.** Cluster ranking of biodiversity and economic growth literature.

| Cluster-ID | Size | Silhouette | Year | Label (LLR) |
|---|---|---|---|---|
| 0 | 76 | 0.904 | 2008 | Conservation planning |
| 1 | 72 | 0.877 | 2013 | Decision-making |
| 2 | 71 | 0.886 | 2010 | Food production |
| 3 | 69 | 0.872 | 2008 | Economic growth |
| 4 | 68 | 0.8 | 2015 | EKC hypothesis |
| 5 | 61 | 0.875 | 2015 | Mitigation banking |
| 6 | 56 | 0.946 | 2009 | Neoliberal conservation |
| 7 | 54 | 0.874 | 2016 | Biodiversity maintenance |
| 8 | 52 | 0.905 | 2011 | Brazil |
| 9 | 46 | 0.967 | 2017 | Ecological footprint |
| 10 | 45 | 0.925 | 2010 | Buffer zones |
| 11 | 44 | 0.909 | 2015 | Forest biodiversity |
| 12 | 43 | 0.833 | 2012 | Ecosystem services |
| 13 | 40 | 0.95 | 2014 | Enrichment planting |
| 15 | 25 | 0.983 | 2008 | Sustainable fisheries |
| 16 | 19 | 0.999 | 2004 | Cross-national |
| 17 | 19 | 0.959 | 2017 | Natural habitat |
| 21 | 10 | 0.998 | 2011 | Biofuels production |
| 24 | 8 | 0.988 | 2017 | Nanotechnologies |
| 30 | 5 | 0.986 | 2013 | Mycorrhizae |

### 3.2. Co-Citation Analysis of Literature on Biodiversity and Economic Growth

In Table 8, the literature co-citation analysis, a total of the top ten most-cited literature are listed. In the Science Journal, a representative one is "Planetary boundaries: Guiding human development on a changing planet" cited 32 times, and "Global Biodiversity: Indicators of recent declines" which was quoted 28 times. The co-citation analysis shows that the top ten articles are mainly concentrated in the middle part of Figure 9, which is also a position closely related to the research of other scholars.

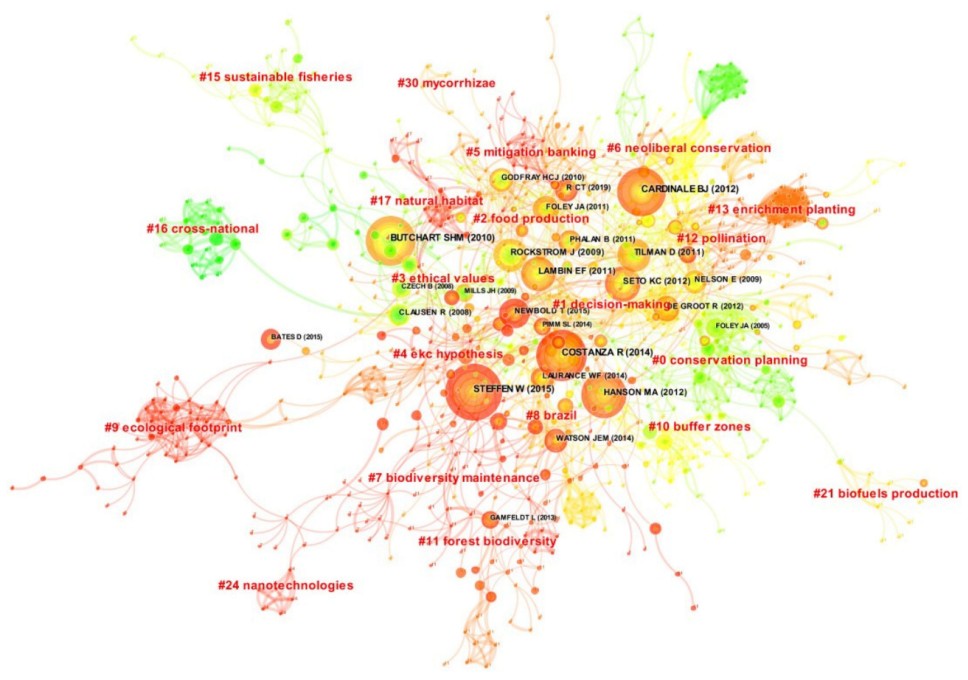

**Figure 9.** Co-citation network structure of biodiversity and economic growth literature [44–53].

**Table 8.** Co-citation rankings of biodiversity and economic growth literature.

| Count | Year | Author | Journal | Title |
|---|---|---|---|---|
| 32 | 2015 | Steffen et al. [44] | Science | Planetary boundaries: Guiding human development on a changing planet |
| 28 | 2010 | Butchart et al. [45] | Science | Global biodiversity: Indicators of recent declines |
| 28 | 2014 | Costanza et al. [46] | Global Environmental Change | Changes in the global value of ecosystem services |
| 28 | 2012 | Cardinale et al. [47] | Nature | Biodiversity loss and its impact on humanity |
| 25 | 2012 | Hanson et al. [48] | Science | Crystal structure of a lipid G protein–coupled receptor |
| 21 | 2012 | Seto et al. [49] | P Natl Acad Sci USA | Global forecasts of urban expansion to 2030 and direct impacts on biodiversity and carbon pools |
| 20 | 2011 | Lambin et al. [50] | P Natl Acad Sci USA | Global land use change, economic globalization, and the looming land scarcity |

**Table 8.** *Cont.*

| Count | Year | Author | Journal | Title |
|:---:|:---:|:---:|:---:|:---:|
| 19 | 2009 | Rockström et al. [51] | Nature | A safe operating space for humanity |
| 19 | 2011 | Tilman et al. [52] | P Natl Acad Sci USA | Global food demand and the sustainable intensification of agriculture |
| 18 | 2015 | Newbold et al. [53] | Nature | Global effects of land use on local terrestrial biodiversity |

*3.3. Burst Detection Analysis of Keyword and Reference*

The burst point detection can reflect the latest research trends in the research field. Strength means the frequency of sudden bursts. The sudden burst here refers to a significant increase in the number of keywords or documents over a period of time. The year represents the time when the keyword or document first appeared. Begin and end represent the start time and end time of keyword or literature emergence. The rectangles in the last column depict the 30 years 1992–2021 and their thin dotted line represents the years that articles have received slight increases in citations while the thickness shows that citations have risen dramatically.

Table 9 presents the important 15 keywords in keyword burst point detection. As shown in Table 9, at the beginning of the 21st century, a large number of relevant keywords were found in the references, such as biodiversity, which has been frequently cited for 10 years. Depending on the keyword burst detection, the most frequently discussed keywords in biodiversity can be divided into two phases. The first stage is before 2000. The research at this time mainly focused on proposing the relevant concepts of biodiversity and building an external research framework. The second stage was the research after 2000, which mainly focused on refining the research direction and fields of biodiversity. Meanwhile, scholars' research has gradually expanded their horizons and related fields have become more diversified.

**Table 9.** Top 15 keywords with the strongest citation bursts.

| Keywords | Year | Strength | Begin | End | 1991–2021 |
|:---:|:---:|:---:|:---:|:---:|:---:|
| conservation | 1991 | 3.5762 | 1995 | 1996 | |
| biodiversity | 1991 | 15.9561 | 1995 | 2005 | |
| diversity | 1991 | 4.1843 | 1996 | 2007 | |
| growth | 1991 | 6.3313 | 1998 | 2006 | |
| sustainable development | 1991 | 3.8971 | 1998 | 2009 | |
| management | 1991 | 3.6765 | 1998 | 2005 | |
| ecosystem | 1991 | 3.7192 | 1999 | 2004 | |
| habitat | 1991 | 4.1171 | 2004 | 2013 | |
| wildlife | 1991 | 3.7056 | 2004 | 2008 | |
| rain forest | 1991 | 3.74 | 2004 | 2006 | |
| timber | 1991 | 3.8028 | 2006 | 2008 | |
| ecology | 1991 | 4.6527 | 2007 | 2011 | |
| extinction | 1991 | 4.6817 | 2007 | 2015 | |
| evolution | 1991 | 3.8637 | 2011 | 2014 | |
| protected area | 1991 | 3.592 | 2012 | 2013 | |

Figure 10 shows the timeline of keywords covered by the 26 years of research. In 1995, the main research topics focused on environmental degradation and the relationship between the ecological environment and human beings. Subsequent research is also based on these keywords as an extension. It can also be seen that the research in the past few years was focused on the practical problems caused by environmental degradation, and the focus in these years has gradually been placed on the economy and the construction of a new way of life.

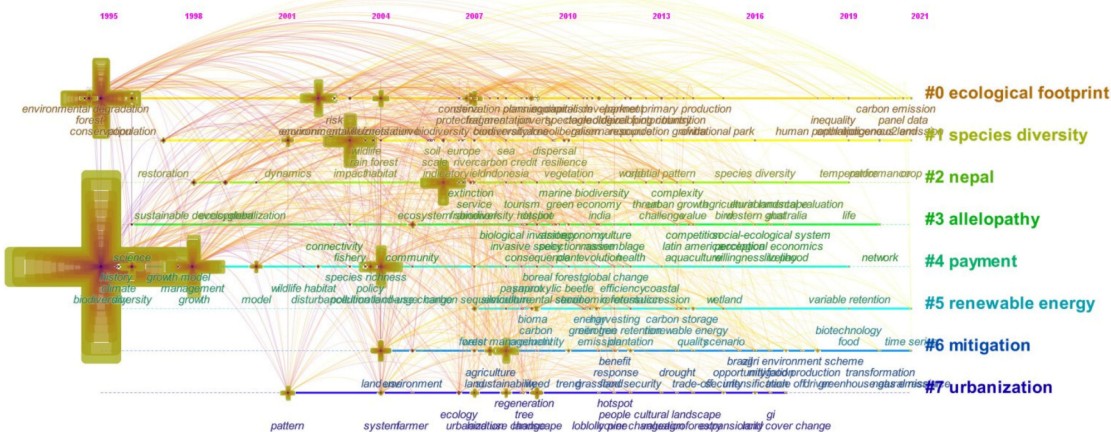

**Figure 10.** Keyword timeline diagram of literature related to biodiversity and economic growth.

The trend of co-citation can be reflected by the analysis of the burst point detection on references. It can be seen that the citations in Table 10, started after 1997, and the most cited literature is "The value of the world's ecosystem services and natural capital".

**Table 10.** Top 10 References with the strongest citation bursts.

| References | Year | Strength | Begin | End | 1990–2021 |
|---|---|---|---|---|---|
| The value of the world's ecosystem services and natural capital | 1997 | 4.1086 | 1998 | 2002 | |
| Biodiversity hotspots for conservation priorities | 2000 | 5.0408 | 2005 | 2008 | |
| Economic associations among causes of species endangerment in the United States | 2000 | 3.8264 | 2007 | 2008 | |
| Modeling joint production of wildlife and timber | 2004 | 4.0815 | 2007 | 2010 | |
| Environmental Kuznets curve: threatened species and spatial effects | 2005 | 3.6967 | 2008 | 2013 | |
| Footprints on the Earth: The environmental consequences of modernity | 2003 | 4.2349 | 2008 | 2010 | |
| The rise and fall of the Environmental Kuznets Curve | 2004 | 4.2789 | 2008 | 2012 | |
| Effects of Economic Prosperity on Numbers of Threatened Species | 2001 | 5.0971 | 2008 | 2009 | |
| Social and environmental influences on endangered species: A cross-national study | 2004 | 4.2349 | 2008 | 2010 | |
| Global biodiversity decline of marine and freshwater fish: A cross-national analysis of economic, demographic, and ecological influences | 2008 | 5.5838 | 2009 | 2013 | |

## 4. Citation Path Analysis of the Relationship Research between Biodiversity and Economic Development

The third part of the article provides a multifaceted analysis of the current development and hotspots of biodiversity research with the help of a bibliometric-based analysis. Based on this, the fourth part will further provide a path analysis of research trends.

This section contains a comprehensive analysis of four main paths, namely, the forward main path, global key main path, global standard main path, and backward main path. Main path analysis is a quantitative and functional method for extracting critical paths through "connectivity". Additionally, it effectively reduces the complexity of knowledge networks. The global standard main path: the most important path extracted from the network is based on the overall importance of nodes in the knowledge flow. The global key main path discovers the critical path with significant impact during the development of the domain. Additionally, the global key main path is to explore other important paths on the global standard main path. The forward main path indicates the flow of knowledge from the past to the present. The backward main path is the opposite of the forward main path, which is a backward path from the present to the past. The forward main path and backward main path help to detect the source of current active thoughts or opinions.

Figure 11 shows the forward main path map of biodiversity and economic growth. Figure 11 shows [5] as the starting point of the study, tracking the trajectory from the past to the present. It can be seen from Figure 11 that the literature [54] is a key node in the forward main path. Based on it, it has developed four branches, namely [11,55–57]. The relationship between economic development and biodiversity has received more attention. However, these research directions have been following the idea of the main pathway. For example, the publication "Economic development, institutions, and biodiversity loss in a global context" provided direction for subsequent research and also generated offshoots. The research directions of this document cover the impact of climate change on biodiversity, environmental protection, environmental pollution, and national differences in the biodiversity crisis. Ref. [56] also extended the analysis of the forward main path.

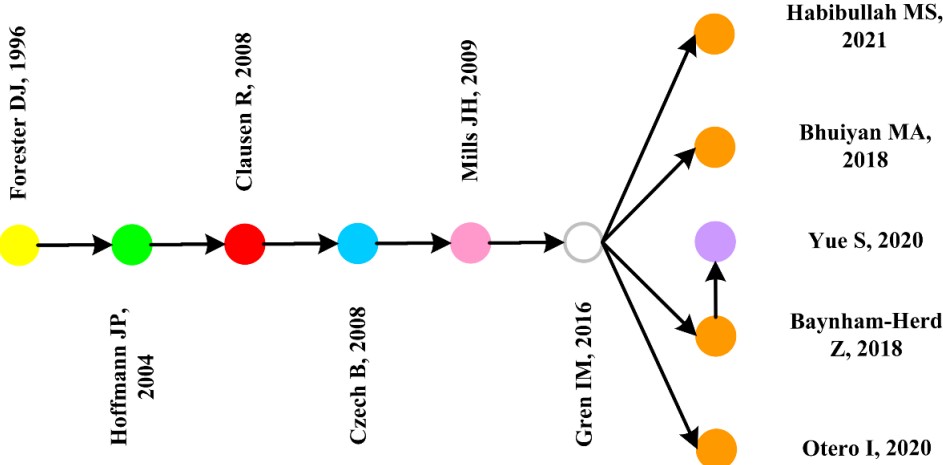

**Figure 11.** Forward main path analysis [5,11,54–62].

A comparison of Figures 12 and 13 reveals significant literature overlap on both pathways. The obvious difference is paper [61], "Effects of economic prosperity on numbers of threatened species," and it forms a branch of the backward main pathway. Figure 13 shows that the development trend shows a straight line, which implies a high heterogeneity of research directions in observing economic development and biodiversity from the perspective of global labeling path analysis.

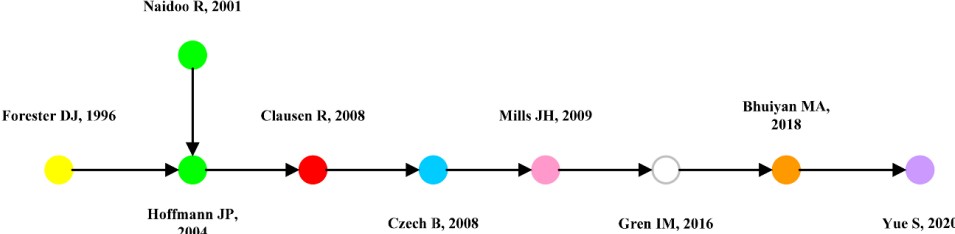

**Figure 12.** Backward main path analysis [5,54,56,58–63].

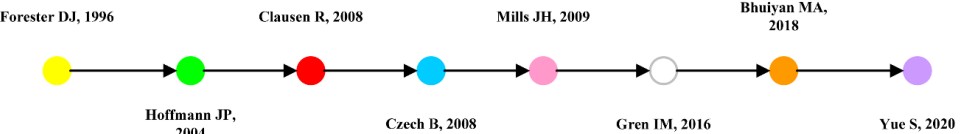

**Figure 13.** Global standard main path analysis [5,54,56,58–60,62,63].

The global key main path is proposed to improve some diffusion paths that have relatively low weights and may be ignored. Therefore, this paper uses the global key main path method to present more detail in a specific area. As can be seen from Figure 14, the previous studies are mainly divided into two trends and, under these two trends, there are also many overlaps and influences.

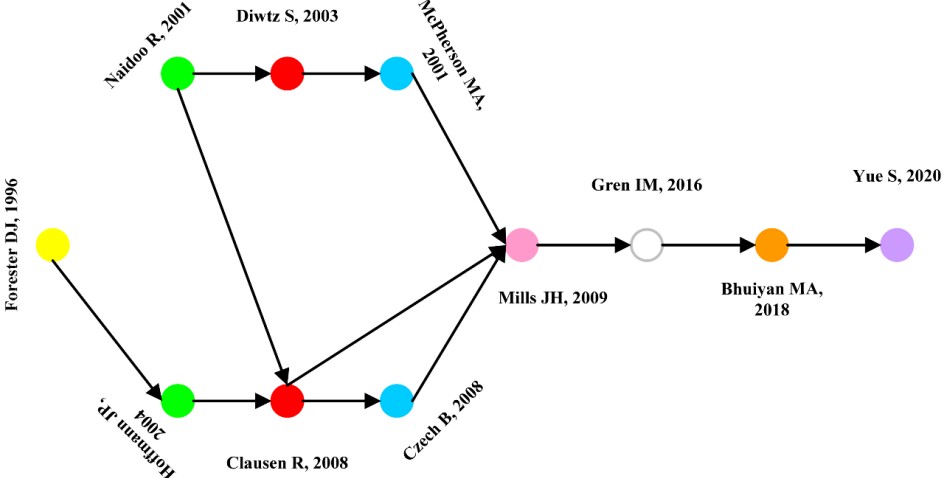

**Figure 14.** Global key main path analysis [5,54,56,58–65].

Through the analysis of these four paths, we found that scholars' influence is very important in the development of the field, for example, Forester and Hoffmann, who are instrumental not only in the number of publications but also in the path nodes. Forester's study, "Modeling human factors that affect the loss of biodiversity", published in 1996 [5], focuses on the relationship between biodiversity change based on human-influenced factors. Hoffman's 2004 study, "Social and environmental influences on endangered species: A cross-national study" analyzed biodiversity change in terms of social factors [58]. The difference between the two studies is that they are grounded in painless influences.

## 5. Conclusions

This paper focuses on research trends in biodiversity and economic growth through bibliometric studies and seeks research hotspots in biodiversity and economic development. We also analyze it from two perspectives. The first is the perspective of bibliometrics. The research status of biodiversity and economic growth, as well as the year, author, institution, country, and keywords that appear in the higher frequency of related literature, have been

studied and analyzed by selecting different node types in the software. After that, the development paths of the literature were carved using Pajek. The analysis of the literature on biodiversity and economic growth was refined by analyzing the forward main path, the global key main path, the global standard main path, and the backward main path.

Finally, the following characteristics can be summarized: (1) based on the descriptive statistical analysis of the literature data, the publication and citation amounts of the literature about biodiversity and economic growth research have increased significantly. (2) By combing and summarizing the classical literature, we find that scholars oppose unrestricted economic growth and advocate for the protection of the environment and biodiversity. Ecological environment protection and economic development are win-win relationships. (3) The keyword analysis revealed that the question of how to develop the economy while preserving ecological diversity is a current research hotspot. (4) Developed countries or regions are pioneers in studying the relationship between biodiversity and economic growth, and they have progressively led and driven the development of research in this field. Although we have demonstrated the bibliometric conclusions of biodiversity and economic growth through these methods, there are still some deficiencies. First of all, words similar to keywords may not be searched comprehensively, resulting in research results, which are problematic in a small probability. Secondly, there are many good bibliometric research methods to make bibliometric conclusions, but we only use two mainstream methods, and in future research, other methods can also be used for research.

**Author Contributions:** Literature review and manuscript writing, Y.Z. The theoretical framework, W.Z. Data collection and processing, D.L. All authors have read and agreed to the published version of the manuscript.

**Funding:** The authors acknowledge the financial support from the Natural Science Foundation of China [72071176]; and Social Science Innovation Team Project of Yunnan Province [No.2022CX01].

**Institutional Review Board Statement:** Ethical review and approval were not required for the study on human participants in accordance with local legislation.

**Informed Consent Statement:** Not applicable.

**Data Availability Statement:** Original contributions presented in this research are contained in the article and further inquiries can be addressed to the corresponding author.

**Conflicts of Interest:** The authors declare no conflict of interest.

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
