# Peer review of "The Relationship Research between Biodiversity Conservation and Economic Growth: From Multi-Level Attempts to Key Development"

_sustainability, doi:10.3390/su15043107_

Round 1

Reviewer 1 Report

Dear Authors, 

Interesting paper.

This study made a bibliometric analysis of biodiversity and economic growth through two tools; CiteSpace and Pajek and claimed that yet most studies have not carried out the bibliometric analysis in this field. This argument seems so ambiguous since the paper cannot show evidence to support the argument (fewer citations). Lines 45 – 89 describe research on biodiversity and economic growth, but these paragraphs cannot directly support the claim. In addition, the introduction section does not seem to contain adequate discussion of the research gap leading to the scientific reasoning of putting this research in place.

Furthermore, related to the discussion section, I found a lack of adequate theoretical support for the arguments has been made. In other words, the discussion mostly showed the data and described the meaning of the data.

The conclusions' content is normative and no identification of the potential limitations and weaknesses for future research to address.

Your paper can be accepted after addressing the above comments.

Thank you and wish you all the best

Reviewer 2 Report

This study conducts a bibliometric analysis between biodiversity conservation and economic growth. Even though the topic is interesting, I think this research has severe flaws:

- The authors do not present the research query used to find the articles.

- There is no mention of data pre-processing and filtering. This step is crucial in bibliometric studies (e.g. check whether there are two different authors with the same name, check whether there are different expressions with the same meaning that will be treated as different knowledge groups in the subsequent software analysis, read the abstracts to assess the relevance of the articles for the field of research). The results may be unreliable and biased if these procedures are not adopted (Gan et al. 2022; Zupic and Cater, 2015).

- The authors do not properly define some terms and expressions (e.g. "Centrality" in Tables 2 to 6, "Silhouette" in Table 7, "Strength" in Tables 9 and 10, "forward main path", "global critical path", "global standard path", and "backward main path" in lines 306 and 307). Thus, it is difficult for a non-specialist in bibliometric studies to fully understand this research without resorting to outside sources.

- The authors do not define the variable that is graphed in the last column of Tables 9 and 10. This makes the results interpretation difficult. 

References

- Gan, Y., Li, D., Robinson, N. and Liu, J. (2022). Practical guidance on bibliometric analysis and mapping knowledge domains methodology – A summary. European Journal of Integrative Medicine 56, 102203.

-Zupic, I. and Cater, T. (2015). Bibliometric Methods in Management and Organization. Organizational Research Methods 18 (3), 429-472.

Reviewer 3 Report

The Manuscript The relationship research between biodiversity conservation and economic growth: from multi-level attempts to key development is fluent and easy to read. I think the present study is valuable. It provides a new perspective to assess research trends in biodiversity and economic growth. It may be accepted for publication in Sustainability .

Author Response

Thank you very much for this positive comment.

Reviewer 4 Report

Dear authors,

I have been invited to review your paper. Attached you will find detailed comments that might help revise it.

In my view, it would be necessary to conduct a fully comprehensive literature review and present a complete state of the art. You might also wish to present more clearly your research questions and hypotheses and introduce clearly the various methodological steps of your research. Finally, it would be useful to put forward policy recommendations stemming from your research findings.

Thank you and regards,

Anonymous reviewer

Round 2

Reviewer 2 Report

This version of the manuscript is better than the previous one. However, I still have several concerns:

- I think concepts such as "Centrality", "Silhouette", "Strength", "Forward main path" and so on are not rigorously defined. If the authors do not want to do it in the main text, I suggest they add an appendix with a detailed definition of all the concepts.

- In lines 132-133, we have "After 2018, there was a marked decrease" (in the number of publications). However, upon observing Figure 1, we notice the opposite is true.

- In lines 184-185, we have "From the Fig.4 institutional collaboration map, it can be seen that countries with more research in this field are concentrated in China". Did the authors mean "From Fig.4 institutional collaboration map, it can be seen that institutions with more research in this field are concentrated in China"?

- The centrality indicator values for Italy and Germany are 0.09 and 0.08 (Table 3). However, in line 205, the authors wrote these values are 0.9 and 0.8.

- The following excerpt (lines 237-242) "The research data starts... of a new way of life" comments on the keywords. However, the authors inserted it into the "Journal and author collaboration networks" subsection. It shouldn't be there.

- The authors claim, in line 271, "The higher the Silhouette scores the same the better the quality". This sentence is confusing. Besides, the authors state, in the note following Table 7, that "Silhouette mainly measures the homogeneity within a cluster after clustering". From these sentences, it becomes difficult to know how to interpret this concept properly.

- I think the definitions of "forward main path", "global critical main path, "global standard path", and "local standard main path" are unclear. The authors do not properly explain the difference between a "global" and a "local" path. Furthermore, (I think) they rename "global critical main path" as "global key main path" in line 315 and use the latter expression afterward. 

- In lines 320-322, the authors wrote "It is evident from the Fig. 11 that the forward main path takes 2016 as the node, generating four branches extending the study to 2020. Which node? The key node? The critical node?

- Looking at Figures 12 and 13, we see that the global critical main path is a superset of the global standard main path. Shouldn't it be the other way around, given the critical path includes only the key nodes?

- There are several typos in the text (e.g. a space is missing in lines 36 and 47 before the parentheses, and the letters "'i" and "a" are superimposed in the name "Díaz", in line 38). Furthermore, there is a lack of uniformity in some expressions that are used throughout the text (e.g. "inverted U-shaped" in line 44 and "inverted u-shaped in line 46). I suggest the authors carefully read the manuscript again to correct these typos and improve the English.

Reviewer 4 Report

Dear authors,

Thank you for having addressed most of my comments. Please check out the English as the text contains some typos. 

Line 30 "biological diversity" (biodiversity) is the term.. Biological should be with B

Line 73 For the above purpose, we analyzes - should read analyze

Thanks and good luck,

Anonymous reviewer

Round 3

Reviewer 2 Report

I would like to thank the authors for taking my comments into account. I think they did a good job. However, I still have two minor concerns:

- I don't fully understand the following passage in lines 347-349: "A comparison of Fig. 13 and Fig.12 reveals significant literature overlap on both pathways. The obvious difference lies in Hoffman's social and environmental impacts on endangered species: A cross-national study from 2004. After comparing the two figures, I found this article is on both paths. Why do the authors claim this is an obvious difference?

- There is an unnumbered figure after Fig. 13, which is equal to Fig. 12. I suppose this is a mistake, and it shouldn't be there.
